# Human Papillomavirus Carcinogenicity and the Need of New Perspectives: Thoughts from a Retrospective Analysis on Human Papillomavirus Outcomes Conducted at the Hospital University of Bari, Apulia, Italy, between 2011 and 2022

**DOI:** 10.3390/diagnostics14090968

**Published:** 2024-05-06

**Authors:** Raffaele Del Prete, Daniela Nesta, Francesco Triggiano, Mara Lorusso, Stefania Garzone, Lorenzo Vitulano, Sofia Denicolò, Francesca Indraccolo, Michele Mastria, Luigi Ronga, Francesco Inchingolo, Sergey K. Aityan, Kieu C. D. Nguyen, Toai Cong Tran, Ciro Gargiulo Isacco, Luigi Santacroce

**Affiliations:** 1Department of Interdisciplinary Medicine (DIM), U.O.C. Microbiology and Virology, University-Hospital of Bari, 70100 Bari, Italy; raffaele.delprete@uniba.it (R.D.P.); d.nesta@studenti.uniba.it (D.N.); francesco.triggiano@uniba.it (F.T.); m.lorusso54@studenti.uniba.it (M.L.); s.gargzone@studenti.uniba.it (S.G.); l.vitulano1@studenti.uniba.it (L.V.); s.denicolo4@studenti.uniba.it (S.D.); francesca.indraccolo@uniba.it (F.I.); m.mastria8@studenti.uniba.it (M.M.); luigi.ronga@uniba.it (L.R.); francesco.inchingolo@uniba.it (F.I.); drkieukaren@gmail.com (K.C.D.N.); luigi.santacroce@uniba.it (L.S.); 2College of Engineering, Northeastern University, 5000 MacArthur Blvd., Oakland, CA 94613, USA; s.aityan@northeastern.edu; 3Department of Basic Medical Sciences and Biomedical Research Center, Pham Ngoc Thach University of Medicine, Ho Chi Minh City 700100, Vietnam; trancongtoai@pnt.edu.vn

**Keywords:** HR-HPV, LR-HPV, cervical cancer, cervical intraepithelial neoplasia, distribution, human papillomavirus, genotypes, adult stem cells, L1 and retrotransposition, polymerase chain reaction (PCR)

## Abstract

Background: The current manuscript’s aim was to determine the human papillomavirus (HPV) genotype-specific prevalence and distribution among individuals, males, and females, of different ages in the region of Apulia, Italy, highlighting the possible variables involved in the carcinogenicity mechanism. In addition, we proposed two hypothetical models of HPV’s molecular dynamics, intending to clarify the impact of prevention and therapeutic strategies, explicitly modeled by recent survey data. Methods: We presented clinical data from 9647 participants tested for either high-risk (HR) or low-risk (LR) HPV at the affiliated Bari Policlinic University Hospital of Bari from 2011 to 2022. HPV DNA detection was performed using nested-polymerase chain reaction (PCR) and multiplex real-time PCR assay. Statistical analysis showed significant associations for all genders and ages and both HR- and LR-HPV types. A major number of significant pairwise associations were detected for the higher-risk types and females and lower-risk types and males. Results: The overall prevalence of HPV was 50.5% (n-4.869) vs. 49.5% (n-4.778) of the study population, of which 74.4% (n-3621) were found to be HPV high-risk (HR-HPV) genotypes and 57.7% (n-2.807) low-risk HPV (LR-HPV) genotypes, of which males were 58% and females 49%; the three most prevalent HR-HPV genotypes were HPV 53 (n707-15%), 16 (n704-14%), and 31 (n589-12%), and for LR-HPV, they were 42 (19%), 6 (16%), and 54 (13%); 56% of patients screened for HPV were ≤ 30 years old, 53% were between 31 and 40 years old, 46% were 41–50 and 51–60 years old, and finally, 44% of subjects were >60 years old. Conclusions: Our study provided comprehensive epidemiological data on HPV prevalence and genotype distribution among 9647 participants, which could serve as a significant reference for clinical practice, and it implied the necessity for more effective screening methods for HPV carcinogenesis covering the use of more specific molecular investigations. Although this is a predominantly descriptive and epidemiological study, the data obtained offer not only a fairly unique trend compared to other studies of different realities and latitudes but also lead us to focus on the HPV infection within two groups of young people and adults and hypothesize the possible involvement of dysbiosis, stem cells, and the retrotransposition mechanism.

## 1. Introduction

Human papillomavirus (HPV) infection is considered one of the most common sexually transmitted diseases worldwide. HPV belongs to DNA viruses of the Papillomaviridae family [1]. HPV’s size is ~8 kb double-stranded DNA, and they are equipped with eight open reading frames (ORFs): E1, E2, E4, E5, E6, E7, L1, and L2, each one executing different tasks; the E1, E2, E4, and E6 maintain viral vital function, whilst L1 and L2 encode structural capsid proteins needed for the virus’s replication [1,2]. The virus is also equipped with two major promoters in charge of regulating the transcription within the HPV genome: (1) the early promoter located in the long control region, which regulates the expression of E6 and E7 and (2) the late promoter located within the E7 gene, in charge of regulating the expression of E1, E2, E4, E5, L1, and L2 [3,4,5,6]. 

Currently, ~200 HPV types are known, categorized into five distinct genealogical groups plus the α-papillomaviruses that include only the HR-HPV and LR-HPV subtypes; this categorization poses significant issues and limitations regarding when we should target global prevalence [7,8,9]. Different variants and subtypes are classified based on their genetic distance from viral genomes, and many are capable of causing a range of mucosal or cutaneous epithelial hyperplastic lesions, which are also considered important causative factors of several types of cancers such as the squamous epithelia of the cervix, oropharynx, and anogenital areas [6,7,8,9]. New HPV subtypes are observed when the L1 ORF shows decidual differences from other known genotypes by ≥10%, while simple variants exhibit between 0.5–1% genome differentiation [10,11,12,13,14]. 

More recent findings in papillomavirus research have highlighted three contributive independent patterns: (i) studies on a rare human genetic makeup highlighted the hereditary predisposition toward some forms of epidermodysplasia and verruciformis, characterized by extensive warts, with the development of warts followed by the devolution of skin cancer with the possible involvement of mature stem cells [15]; (ii) data from the determination of different pathogen’s etiology of cancer suggested the possible role of gut/vaginal dysbiosis [16]; (iii) immune compromission and, therefore, the importance of uncontrolled autoimmune inflammatory responses have been highlighted [17].

### The HPV Epidemiology and the Different Infectivity Pathways

Approximately 15–20% of cancers worldwide are associated with specific viruses known to be implicated in the development of cancer, including HPV, Epstein–Barr virus (EBV), Hepatitis B virus (HBV), and hepatitis C virus (HCV), which are known to be the most popular. Demographically and epidemiologically, the outcomes have shown that in sexually active men and women, the HPV infection rate could reach 80% [6,7,8,9,10,11,12,13]. Most of the time, the infections tend to resolve spontaneously, due to the healthy host’s immune responses; only in a very few cases, the virus persists and eventually triggers carcinogenesis [14,15,16,17,18]. Recent studies have shown the numerous and highly sophisticated mechanisms adopted by these viruses to evade or suppress immune responses [18,19,20,21,22]. 

As obligate intracellular “parasites”, these viruses encode proteins that affect cell development, the growth cycle, and the apoptosis mechanism. Within the host cells, each virus regulates its development needs by reprogramming the host’s cell signaling mechanism, disrupting major checkpoints in charge of regulating proliferation, differentiation, and genomic integrity, especially through deactivation of the interferon mechanism (IFN) through sequestering STAT-1 and -2 [18,19,20,21,22]. Several variables such as local and systemic immunity deficiencies, immunosuppression, genetic predisposition, and environmental factors (pro-carcinogenic exposure) may contribute to constant silent mutation as a result of chronic viral infection [23,24,25,26,27]. 

HPV E6 and E7 are crucial oncogenes that promote HPV carcinogenesis that easily occurs in combination with HR-HPV types such as 16 and 18 [25,26,27,28]. Within the basal epithelium, HPV begins the cellular sabotage phase by “reprogramming” the host cell machinery to continue replication; the next step is characterized by the release of HPV from terminally differentiated cells that detach from the epithelial surface [29,30,31,32,33]. Anti-HPV antibodies are only detected 6–12 months after infection; during this period, HPV adopts a number of evasion mechanisms that can be particularly dangerous for immunocompromised hosts [25,26,27,28].

Healthy immunosurveillance plays a crucial role in initiating antigen-dependent responses against HPV infection and virally transformed cells. For example, numerous studies have confirmed that 90% of anogenital HPV infections are substantially eliminated in immunocompetent hosts [28,29,30,31,32,33]. It should be noted that intracellular HPV is paradoxically protected by the host’s immunity, which allows the virus to reach maturity in order to exit, first infecting the keratinocytes and then exfoliating; throughout this process, the immune response remains relatively low compared to immune responses to viruses that are not confined to epithelial tissues [32,33,34,35]. 

Genetic predispositions linked to HPV-related disease may include errors or polymorphisms of CIB-1, CXCR-4, EVER-1, EVER-2, DOCK-8, and GATA-2 genes, since a small number of critical mutations in these genes have been demonstrated to trigger deep skin HPV infections, linked to being highly susceptible to CC [25,36,37,38,39]. People living with secondary immunodeficiency due to solid-organ transplants and acquired immunodeficiency (human immunodeficiency virus (HIV) infection) are at significant risk of HPV-related infection. They are exposed to cervical, anogenital, and oropharyngeal carcinomas [40,41,42,43,44]. 

Dendritic cells (DCs), natural killer (NK) cells, macrophages, and Langerhans cells (LCs) are crucial during the infection’s initial phases, while HPV-specific CD8 + T cells start targeting early viral proteins exposed by infected cells [44,45,46,47]. In HPV infection immune responses are mainly linked to a class II MHC that enhances T cell activation toward TH2-like regulatory T cell (Treg) function [44,45,46]. From this point, these regulatory cells start producing several cytokines and chemokines that become the specific trait of patients with severe recurrent respiratory papillomatosis (RRP) [44,45,46]. These patients with RRP have shown impaired NKs and macrophages. While HPV inhibits the natural killer cell immunoglobulin-like receptor (KIR) gene haplotypes [43,44,45], HPV also affects macrophage translocation during acute infection by interfering with macrophage chemotaxis, reducing the number of proinflammatory cytokines, such as TNF-α and IL-17A, both important in the recruitment of immune cells at the site of infection [43,44,45]. In addition, HPV is able to make macrophages a reliable partner for tumor progression using the E6 oncoproteins, which are involved in the inhibition of monocyte chemotactic protein MCP-1 from infected keratinocytes, once the malignant transformation is settled [42,44]. 

On the other hand, HR-HPV 16 oncoproteins E6 and E7 have been shown to downregulate the secretion of macrophage inflammatory protein 1α (MIP1α) from infected keratinocytes, which in turn, impairs the immune response by promoting the production of MIP1α, transforming growth β (TGFβ) by increasing the activity of the TGF-β1 promoter [45,46,47]. Consequently, an increased level of TGF-β expression has been observed in HPV-derived tumors, which tends to induce local immune suppression. This, in turn, increases susceptibility to malignant transformation [47,48,49]. In an analysis of human tissue specimens and cultured keratinocytes, scientists found that the HR-HPV E7 protein was involved in the hypermethylation of the CXCL14 promoter. Of note, the CXCL14 is a chemokine expressed only in normal cells, that cannot be found in mutating and cancerous cells [45,46,47,48,49]. 

Interestingly, experiments conducted on animal models with head/neck and cervical cancers revealed that restoring CXCL14 expression in HPV-positive cancer cells showed an evident increase in NK and T cells in the tumor-draining lymph nodes, contributing to strong anti-tumor responses in vivo [50,51,52]. Outcomes have demonstrated the presence of CXCL14-producing fibroblasts with maturing macrophages, which also implicitly refers to a possible role for CXCL14 in macrophage development [53,54,55,56,57,58]. Thus, it is not surprising that HPV has evolved a mechanism capable of affecting these chemokines that contribute to its immune escape strategy [59].

Pathophysiologically, the HPV integration into the host genome indicates the precise moment at which oncoproteins E6 and E7 start activating; both are, respectively, associated with p53 and pRb degradation, which in turn induce the increase in the proliferation rate, tumor growth-promoting transcription factors with a consequent increase in enzyme gene expression important in replication and cell division [57,58,59]. Similarly to what happens in other obligate intracellular pathogens, HPV, through E7, interferes during the mitosis process causing mitotic anomalies, favoring the creation of misaligned or delayed chromosomes and breaks in the chromosomal structure, leading to further destabilization of the cellular genome [59,60,61]. In addition, HPV infection of one genotype may enhance the risk of being infected by a new HPV genotype on follow-up [59,60,61]. 

This study covers eleven years of routine diagnostic data on HPV infection and aims to retrospectively evaluate the impact of HPV prevalence and HPV genotype distribution on samples collected from patients in the Apulia region. To address this issue, a quasi-experimental and theoretical approach was conducted to evaluate the effect of real-time PCR on HPV prevalence and the prevalence of individual viral genotypes and to discuss those mechanisms that may facilitate HPV infection to lead to carcinogenesis in youth, adults, and the elderly.

## 2. Material and Methods

### 2.1. Samples Collection

In the period between 2011 and 2022 in the Laboratory of Molecular Biology, UOC Microbiology and Virology, Hospital University of Bari, (Apulia region, Southern Italy), 9647 biological samples were analyzed from patients admitted to the same hospital and from outpatients or during follow-ups. Cervical, vaginal, urethral, and mouth swabs were collected by a rigid cotton-tipped swab applicator (Nuova Aptaca, Cannelli, Italy), which were treated with 2 mL of phosphate-buffered saline (pH 7.4) (Sigma-Aldrich, Milano, Italy); other types of swabs (e.g., urethral) were treated with 1 mL of saline; then they were vortexed for 30 s to favor the detachment of the cells and the genetic material present. Bioptic samples were finely chopped and placed in Green Beads tubes with 1 mL of DNA Lysis Tissue Buffer (Roche, Milano, Italy); the samples were then removed and homogenized with three cycles with DNA Magna Lyser (Roche, Milano, Italy). The final product was moved into a new tube with 20 ul of Proteinase K (Merck Group, Italy).

### 2.2. DNA Extraction and Multiplex Real-Time PCR

Extracted DNA samples were submitted to multiplex real-time PCR (mRT-PCR) by an Anyplex™ II HPV 28 Detection System (Seegene, Seoul, Republic of Korea) performed on a CFX96 Real-Time PCR (Bio-Rad, Hercules, CA, USA). We started with 200 µL of sample DNA extraction, and the DNA was eluated in a final volume of 100 µL. The procedure includes the first round of amplification of the L1 region viral genome, and the following amplification includes biotinylated primers. PCR products are tested using 3% agarose gel electrophoresis with ethidium bromide to display DNA under ultraviolet light. The PCR products were typed by using a Reverse Line Blot Hybridization Ampliquality HPV-Type Kit (AB Analitica, Padova, Italy). The Ampliquality HPV-Type Kit allows for the detection of 14 HR-HPV types (types 16, 18, 31, 33, 35, 39, 45, 51, 52, 56, 58, 59, 66, and 68), 5 IIB HPV genotypes possibly oncogenic (26, 53, 69, 73, and 82), and 11 LR-HPV types (types 6, 11, 40, 42, 43, 44, 54, 61, 70, 72, and 81) [62].

The first amplification procedure was used to assess the suitability of the extracted DNA, from the thiosulfate sulfurtransferase (TST) gene region (202 bp). The results were assessed by following the manufacturer’s procedures; therefore, the negative TST amplification result underlines the presence of PCR inhibitors or degraded DNA. Moreover, the Ampliquality HPV-HS Bio Kit provides either a negative control or one positive control represented by plasmid DNA of HPV-54.

The current kit allowed the detection of one negative control and three positive controls for each of the two PCR reactions (panels A and B). Panel A includes 14 HR/HPV types, while panel B includes 5 HPV genotypes, IIB, and nine LR types. Data recording and interpretation were automated using the Seegene viewer software, according to the manufacturer’s instructions.

### 2.3. Statistical Analysis

Data and outcomes were performed using SPSS (version 19.0) and WPS (version 2022). We used a descriptive statistical analysis to highlight and confirm HPV prevalence and genotype distribution. SPSS 19.0 for Windows (SPSS Inc., IL, USA) was used to determine single, double, and multiple HPV infections correlated to infections with single or multiple HPV genotypes. Results of the prevalence in the designated groups were assigned corresponding 95% confidence intervals (95% CI). Data and results among gender, age groups, and categorical variables were calculated by using the chi-square test, while a linear-by-linear association test and gamma values were used to assess changes in HR-HPV and LR-HPV prevalence across the different groups. Results were considered statistically significant at *p* values less than 0.05.

## 3. Results 

In the period between 2011 and 2022, 9647 biological samples were analyzed from patients admitted to the same hospital and from outpatients or in follow-ups. Of these, 4869 (50.5%) tested positive, while 4778 (49.5%) were negative (*p* value > 0.05, 95% CI, 95%), of which 74.4% (n-3621) were found to be HPV high-risk (HR-HPV) genotypes, and 57.7% (n-2807) low-risk HPV (LR-HPV) genotypes; this was statistically significant (*p* value < 0.05, 95% CI, 95%) (Figure 1 and Figure 2). In particular, 764/1318 (58%) male subjects tested positive, while 4105/8329 (49%) tested positive for women (*p* value < 0.05, 95% CI). The difference in prevalence between females and males was statistically significant (*p* value < 0.05, 95% CI). There was a lower percentage of males versus females infected with high-risk genotypes (HR-HPV females 45% vs. males 27%), with a greater percentage of males than females with low-risk genotypes (LR-HPV males 42% vs. females 23%) (*p* value < 0.05, 95% CI). Equal percentages of males and females were infected with both risk genotypes (32% vs. 32%) (*p* value > 0.05, 95% CI, 95%). The age difference between female and male patients was statistically significant (*p* value < 0.05, 95% CI) (Table 1). The three most prevalent HR-HPV genotypes were HPV 53 (n707-15%, 95% CI), 16 (n704-14%, 95% CI), and 31 (n589-12%, 95% CI), and for LR-HPV, they were 42 (19%, 95% CI), 6 (16%, 95% CI), and 54 (13%, 95% CI); 56% of patients screened for HPV were ≤30 years old (95% CI), 53% were between 31 and 40 years old (95% CI), 46% were 41–50 and 51–60 years old (95% CI), and finally, 44% of subjects were >60 years old; this difference was statistically significant (*p* value > 0.05, 95% CI, 95%).

Of these, 32% were positive for high- and low-risk genotypes, 42.3% of positive subjects had only high-risk genotypes, and finally, 25.6% of subjects had only low-risk genotypes (Figure 3).

Among the high-risk genotypes detected, we have the following: HPV 53 (15%), HPV 16 (14%), HPV 31 (12%), HPV 66 (9%), HPV 51, 58, 68 (8%), HPV 52 and 56 (7%), HPV 59 (6%), HPV 18, 39, 73 (5%), HPV 33 (4%), HPV 35 and 45 (3%), HPV 82 (2%), and finally, HPV 26 and 639 (0.3% and 0.2%, respectively) (*p* values < 0.05, 95% CI, 95%) (Figure 3).

Table 2 shows the values with relative percentages of subjects (males and females) positive for the HPV test with distinction by genotypes (high, low, both).

However, among the low-risk genotypes detected, we have the following: HPV 42 (19%), HPV 6 (16%), HPV 54 (13%), HPV 44 and 61 (7%), HPV 40, 43, 70 (4%), HPV 11 (3%), and finally, HPV 81 and 72 (0.4% and 0.02%, respectively) (*p* values < 0.05, 95% CI, 95%) (Figure 4).

Furthermore, it was shown that 56% of patients screened for HPV were ≤30 years old, 53% were between 31 and 40 years old, 46% were 41–50 and 51–60 years old, and finally, 44% of subjects were >60 years old (*p* values < 0.05, 95% CI, 95%) (Figure 5).

Over the years covered by the study, the number of people tested for the presence of HPV varied. In particular, from 2011 to 2019, the number of people subjected to the test increased more and more, reaching a quota of 1285 subjects in 2019. In 2020, on the other hand, the number of subjects decreased, remaining at around 1000 subjects (Figure 6).

## 4. Discussion

The present study investigated the presence and spread of HPV genotypes in a hospital in the Apulia region over 11 years. Results showed that a smaller proportion of males than females were infected with high-risk genotypes, while a greater proportion of males than females were infected with low-risk genotypes. Overall, and in line with previous studies, significant differences in HPV infection rates were found among the age groups and genders (Figure 2, Figure 3 and Figure 4). 

During the period covered by this study, the number of people tested for the presence of HPV varied. In particular, from 2011 to 2019, the number of people subjected to the test increased more and more, reaching a quota of 1285 subjects in 2019. In 2020, on the other hand, the number of subjects decreased, remaining at around 1000 subjects (Figure 5, Figure 6 and Figure 7). In addition, as reported by our study conducted in 2019, our data showed that the 53 genotype was prevalent, while the 53-16 and 31 HR genotypes were seen correlated with 42-6 and 54 LR-HPV genotypes and remain the most representative HPV types involved in co-infections in the Bari province. The outcomes showed a higher prevalence of HR-HPV infection in women than men, while there was a higher prevalence of LR-HPV infection in males.

The prevalence of HPV infection was almost equally distributed between younger and elderly people; as reported elsewhere, we have detected the most infected groups within the ages 31–40 (tot n-2.695) and 41–50 (tot n-2.672) [63,64]. Such data confirm the need for a strict screening of the general population (male, female, young adult, and elderly) as a viral reservoir, which eventually explains a constant risk of virus-related pathologies such as genital warts, penile intraepithelial neoplasia, and CC [32,62,63]. A high prevalence of HR-HPV infections was detected. Besides social behavior due to poor sexual campaigns among young adult people on HPV infection-related problems, we proposed something quite new; findings from this study showed that 56% of HPV-affected individuals were ≤ 30 years old, and 53% were between 31 and 40 years old, of which 23% were males and 45% were females, all affected with HR-HPV. This equal division between young and adult/elderly people may hide something interesting indeed that we propose as a possible hypothesis in the following sections.

### 4.1. The Possible Role of Retrotransposition-L1 and Local Stem Cells in Increasing HPV Proliferative Capacity in Young Population Infection

Obligate intracellular pathogens such as HPV and *C. trachomatis* have a severely degraded viral genome; they are not able to completely generate their own energy and metabolic needs and must necessarily depend totally on the metabolic functions of the host cell for their survival [63,64]. However, not all cells may be functional for these pathogens. Therefore, it seems that the latter prefer cells with specific traits that better adapt to their needs. Papillomaviruses and stem cells are a topic that has been proposed by a few important studies in which adult stem cells seem to play a key role in the viral life cycle, particularly during maintenance, and virus-route carcinogenesis [63,64]. We proposed that both HPV and *C. trachomatis* are capable of using local adult stem cells, because of their plasticity, to establish the infection within the vagina and urogenital system [65,66]. 

The 50% of HPV infections among young adults revealed by our study might be linked to the presence of a greater number of stem cells in precancerous and malignant cervical cultures in the range of age. Viral oncogenes have been shown to modify stem cell dynamics, cellular stemness, and neighbor cells, and previous studies have shown a direct interplay of the stem cell niches practically located in every tissue with the extracellular matrix [67,68,69,70,71]. Interestingly, findings from recent studies revealed new information coming from infection biology and stem cell biology, which showed a developmental reprogramming of certain lines of committed mature cells to progenitor stem cell-like cells as a result of intracellular intruders, such as Mycobacterium leprae, *C. trachomatis*, or HPV. This would imply the acquisition of new differentiation and immunomodulation traits by such reprogrammed cells that add particular advantages for obligatory intracellular pathogen spread [69]. This is a sophisticated and often forgotten cellular manipulation mechanism whereby pathogens hijack the genomic plasticity of local mature stem cells by acting on the expression of Sox10 and L1 retrotransposition [69].

Of note, in HPV persistent infections are crucial to the presence of basal cells with similar features to stem cells with a high self-reproductive rate, proliferative ability, and plasticity, which constitute the ideal environment for HPV intracellular settlement. Epithelial transition within the endo-ectocervical and anorectal junctions are characterized by a high presence of mature somatic stem cells and multipotent mesenchymal stem cells and are the anatomical sites most susceptible to carcinogenesis due to HR-HPV aggression [69,72,73,74]. These instances are confirmed by several outcomes that showed the presence of viral genomes within these anatomical locations, which are consistent with those of stem cells as the preferred sites of HPV maintenance or cancer-initiating cells [69,72,73,74]. Thus, by selecting local adult stem cells as niches, HPV and *C. trachomatis* have acquired wide advantages for their development, proliferation, and expansion [69,72,73,74]. 

A second aspect that is often underestimated is shown by biopsies of invasive cervical carcinoma (squamous cell carcinomas and adenocarcinomas) with genotyped HPV, in which results confirmed the presence of a large quantity of expressed retroelements (L1) correlated with DNA methyltransferase 1 expression [69,72,73]. L1 retrotranspositions are known to take place during early human embryonic development and, therefore, along the totipotent lineage of embryonic stem cell differentiation [69]. They have been identified in different progenitor adult stem cell lineages and have been observed in various tissue cell lines in experiments performed to target L1 elements as well [72,73,74]. Interestingly, among uterine differentiated tissue cells, local adult stem cells, which derive from the niche of multipotent adult precursors reveal remarkable plasticity, confirmed by the ability of adult stem cells to switch between differentiated and de-differentiated states following cervix and endometrium injuries [73,74]. Endogenous DNA retroelements are an integral part of the human genome and are key factors in the evolution mechanism of human eukaryotic cells [69,72,73]. However, these elements have been demonstrated to have a role in tumorigenesis and cancer progression as well. Several studies have reported their expression as biomarkers and immunotherapeutic targets for cancer [72,73,74,75]. Outcomes obtained from RT-PCR analysis confirmed the hypomethylation of L1 in either squamous cell carcinoma or carcinoma in situ from the uterine cervix, confirming a close correlation between the two events: the more the hypomethylation, the higher the cancer progression [68,69,70,71]. Therefore, the LINE-1 methylation status may be used as a non-invasive early diagnosis in women at risk of cervical cancer since several lines of evidence showed a higher percentage of differential L1 expression in cancerous and precancerous samples with HPV co-infection compared to normal tissues [73,74,75].

This demonstrates HPV’s ability to affect gene expression through random integration in the human genome either in normal or tumor stem cells. In cervical mutation, HR-HPV 16 and HPV 18 are the main variants responsible as they hijack the LINE-1 Sox10 of host stem cells, which leads to hypomethylation and is crucial in cervical cancer progression [74,75,76]. The HPV strategy within the nuclear compartment is the Sox10 removal accompanied by silencing Sox10 genes that are deeply correlated to the DNA methylation of the Sox10 promoter region in reprogrammed cells, results that highlighted the effective role of the L1 retroelement epigenetic regulation of Sox10 induced by intracellular HPV [69,72,73,74,75,76].

Intriguingly, it was observed that a high number of retroelements were found differentially expressed between tumor histological types and between HPV types, including several HERV families (HERV-K, HERV-H, HERV-E, HERV-I, and HERV-L) [74,75,76]. Recent outcomes showed the highest proportion of differentially expressed L1 elements between HPV mono- and co-infections. Of note, three HERVs and seven L1 were seen extremely close to IL-20 family signaling genes (IL 19, IL 20, IL 20 RA, IL20RB, IL22RA2), and L1 was seen as capable of affecting IL-20 family signaling gene expression [74,75,76]. Eventually, IL20RB overexpression has been correlated with a high rate of cell invasion and migration, a higher grade of cell proliferation, and a poor prognosis in different types of carcinoma, especially that of the kidney [77,78]. Overall, the outcomes were indicative of a combined action between L1 and IL20 family gene expressions and their final contribution to cancer development [74,75,76,77,78]. 

A possible clarification may be obtained by analyzing the specific behavior of Schwann cells versus potential injuries. In response to injury-induced signaling, Schwann cells have been shown to be capable of switching off the myelination program following the loss of axonal contact to acquire a phenotype resembling immature cell status. Therefore, in a similar way, local cervix endothelial and stromal cells, in response to HPV invasion, may re-enter the cell cycle and de-differentiate, to differentiate again once HPV is ready to enter the successive phases [79,80,81]. Animal models confirmed these clinical findings and showed the way the sub-populations of infected hair follicle cells increased their clonogenic activity, which is a typical feature of adult tissue stem cells [79,80,81]. 

Notably, in the presence of cutaneous lesions due to HPV or derived carcinoma, unsorted Lrig1+ cells (human interfollicular epidermal stem cells) demonstrated an increased colony-forming efficiency consistent with an expansion in keratinocyte stem cell numbers [69]. This result infers papilloma as a possible result of continued keratinocyte stem cell expansion into the adjacent overlying epidermis, the same as in human benign viral warts as a result of HPV-infected keratinocyte adult stem cell expansion [69,71,72,73].

### 4.2. HPV Relationship with Different Pathogens and the Mutual Benefits

Recently, it has been confirmed that about 70% of cervical, lung, and oropharyngeal cancers may be caused by HPV, of which the 16 and 18 genotypes are considered the main genotypes responsible [82,83,84,85]. Two main mechanisms were proposed to be involved: (i) the co-presence of bacteria, fungi, and other pathogens that may induce an epithelial disruption, facilitating HPV’s entry into basal keratinocytes of the mucosal epithelium by blocking the immune response necessary for virus clearance and (ii) the presence of an already existing immunocompromised system (as above explained) [83,84,85,86,87]. The presence of E6/E7 oncogenic proteins of HR-HPV genotypes was reported to promote the chlamydial development cycle in the host cells [83,84,85,86,87]. Initially, a low copy number of the HPV genome is kept in “standby” mode, and following epithelial cell differentiation with ongoing chlamydial infection, the HPV commences to replicate faster with higher copies expressing the capsid genes (L1 and L2), which generates new progeny virions that are released from the epithelial surface [84,85,86,87]. 

The mutual benefits between HPV and bacteria like *C. Trachomatis* take place thanks to specific mechanisms that regulate the proliferation and persistence phases within the host cells; the presence of vacuoles surrounded by a membrane called “inclusion” in *C. Trachomatis* allows the replication immediately followed by the transition from EB (elementary body) to RB (reticulate body) and back, until the formation of the final form that indicates the right time to exit the host cell (extrusion and lysis) to start infecting other cells [87,88,89,90]. From this stage, the *C. trachomatis* commence enhancing the oncogenic pathway of Ras–Raf–MEK–ERK with the production of ROS, generating the ideal microenvironment to support not only mutated cell growth but HPV as well [87,88,89,90]. As previously mentioned, bacteria like *C. trachomatis* are also able to create mitosis spindle defects during the host cell’s replication mechanism by avoiding the spindle assembly checkpoint (SAC) [89,90]. The cell is forced to conclude prematurely the mitosis, losing the capacity to perform the right corrections. At this point, both HPV and *C. trachomatis* keep growing and developing within the host cells indefinitely [89,90]. An additional event adopted by *C. trachomatis* that facilitates HPV’s steady development is determined by the subversion cell’s histones obtained by the upregulation of PH2AX and H3K9me3, both hallmarks of DNA double-stranded breaks (DSBs) and linked to senescence-associated heterochromatin foci (SAHF); this is extremely beneficial for HPV’s safe growth [91,92,93,94].

### 4.3. The Role of Microbiota in the Carcinogenesis Process Induced by HPV in Adults and the Elderly

As expected, once our data was compared with other countries and latitudes, there were important differences. As observed by Przybylsk and colleagues in their metanalysis, data need to be evaluated, keeping an eye on different screening procedures, latitudes, environments, and social and food habits that may have influences either on the HPV genotype itself or the microbiota composition [95,96]. For instance, a study group from Russia was exclusively focused only on cytology and included healthy women as well as those with pathology [95,96,97,98]. In addition, even data related to the most common HPV genotypes are quite divergent. For us, the most common, though with minimal percentage difference, was the HPV 53 genotype followed by the 16 and 31 genotypes. On the other hand, the first and second most frequently observed in Spain, Canada, and France were 16 and 31, while in China, Russia, and New Zealand, the most frequent were 16 and 52, in Venezuela, 16 and 18, in Mexico, 16 and 51, and in Portugal, 16 and 58 [95,96,97,98]. Of note, HPV 53 is defined as a “probable high-risk type”, and it is starting to be recognized as one of the four “emergent” genotypes, with a possible role in oncogenesis [98,99]. Therefore, the frequency of different genotypes might be linked to different latitudes and different social and behavioral habits and thus have a CC screening based on different genotypes [95,96,97,98].

The elderly individuals included in this study accounted for 46% (aged 41–50 and 51–60 years) and 44% (>60 years); to explain these data, we proposed the involvement of dysbiosis as the most important cofactor for the high incidence of HPV in this age group [98,99,100]. In this group, the most affected by HR-HPV genotypes were females. Aging contributes together with diet to vaginal microbiota changes that are critical for HPV acquisition, persistence, and clearance [100,101,102]. Dysbiosis in microbiota was observed to promote infection with sexually transmitted pathogens, and HPV infection can also increase the bacterial diversity in both the vagina and male reproductive tract, increasing the chances of contracting CC [100,101,102,103,104]. 

The presence of specific lactic acid-producing bacteria within the vagina and ectocervix microenvironments are important lubricant factors that trap invading pathogens [105,106,107,108,109]. These valuable bacteria are inextricably linked to the acidity of the vaginal environment, crucial for the microbiome’s homeostasis, maintaining a pH of 4–4.5 [105,106,107,108,109]. Therefore, in the vagina, a pH of 4.5 or below serves as a protective shield against harmful pathogens that are unable to survive in such an acidic environment. On the other hand, an alkaline pH favors harmful bacteria gaining the opportunity to move in the damaging vaginal ecosystems [105,106,107].

The vaginal microbiome composition is differently expressed over the reproductive tract, and *Lactobacillus* spp. appears to be the majority in the uterus, while the non-*Lactobacillus* spp. are the major communities of the uterine cervix. In the vagina, the presence of *Lactobacillus* spp. of the Community State Types (CST) I, II, III, and V was observed; the CST-IV is composed of different anaerobes groups, indicating that a predominance of CST-IV is clinically related to bacterial vaginosis [105,106,107]. 

Furthermore, the vaginal microbiome exerts an endocrine function and represents a crucial aspect, and through a complex communication pathway, it is linked to the endocrine glands producing the right amount of the needed hormones, such as estradiol (E2), serotonin, prolactin, and testosterone [107,108,109]. This relationship is multilevel and bidirectional, and the host flora depends on sex and hormones via mutual interactions between its metabolites, the immune system, inflammatory patterns, and nerve–endocrine/paracrine pathways [107,108,109]. For instance, the involvement of E2, serotonin, and prolactin with pathogens such as HPV and C. trachomatis in carcinogenesis is well-established not only in CC but in breast cancer as well [106]. In breast carcinoma, findings have determined estrogen-induced cell proliferation and increased DNA double-strand breaks (DSBs) with BRCA mutations that impair DSB repair due to HR-HPV infection as well [106]. In an E2 deficit, HR-HPV oncogenes E6 and E7 tend to impair the hormone receptor (Hr) pathway by reducing the mobilization of the HR protein RAD51 (pivotal enzymes for DNA-DSB repair by the HR pathway), and the results indicated that E2 could be involved in the carcinogenesis of CC due to HR-HPV infection as well, due to HPV interference on the HR pathway [106].

Therefore, aging has a big impact on the vaginal microbiome composition, since it is deeply altered during and after menopause due to abnormal levels of hormones [105]. Generally, the vaginal microbiota includes *Gardnerella vaginalis*, *Ureaplasma urealyticum*, *Candida albicans*, and *Prevotella* spp., as minor commensals; during the aging process, the prevalence of *Lactobacillus* progressively declines [102,103]. The decrease in lactic acid bacteria in menopausal women is also suggestive of immune-biological and physiological changes, which tend to affect the normal vaginal microbiome, the main cause of vulvovaginal atrophy and vaginal dryness, promoting the settlement and development of aggressive pathogens and HPV [107,108,109,110].

The strength of these outcomes is that the entire issue should be stratified by age to better interpret the vaginal microbial compositions and hormone loading of HPV-positive women with different stages of cervical lesions [107,108,109,110]. Although no one could conclude a definitive causal relationship between the vaginal microbiome, hormone level, and HPV viral load with cancerous development, nevertheless, there is a strong necessity to conduct longitudinal studies to better understand these relationships and the progression of cells toward anomalous mutation [107,108,109,110].

## 5. Strengths and Limitations

To date, it is rare to see the HPV 53 genotype in the first place among such a vast number of individuals. In addition, the analysis of evidence-based medicine supported by this study and the included literature provide a higher-quality foundation for the next clinical practice [70]. Our outcomes are intended to highlight the attention of clinicians providing valuable recommendations for prevention, treatment, and prophylaxis.

Our study has several limitations. First of all, in this study, the overall data of the Apulian population have not been reported but only those relating to a hospital that carries out this type of investigation; therefore, the results cannot be generalized. In addition, no cytological or pre-existing disease data were reported for individuals enrolled in the study. To this end, since such data were not available, it was not possible to correlate the presence of HPV with the cytological and/or histological ultimate findings. Furthermore, the lack of clinical and behavioral information on individuals did not allow us to better characterize the analyzed sample population for the roles of these exposures, which should be considered key variables in the spread of HPV infections; therefore, we were unable to define the association between these variables and the prevalence of HPV.

## 6. Conclusions

To conclude, there is a need to collect more explicative information about the covariables that eventually affect the vaginal microbiota or urogenital tract, such as the status of the genetic makeup, metabolic disorders, smoking, and hormonal contraceptive use.

## Figures and Tables

**Figure 1 diagnostics-14-00968-f001:**
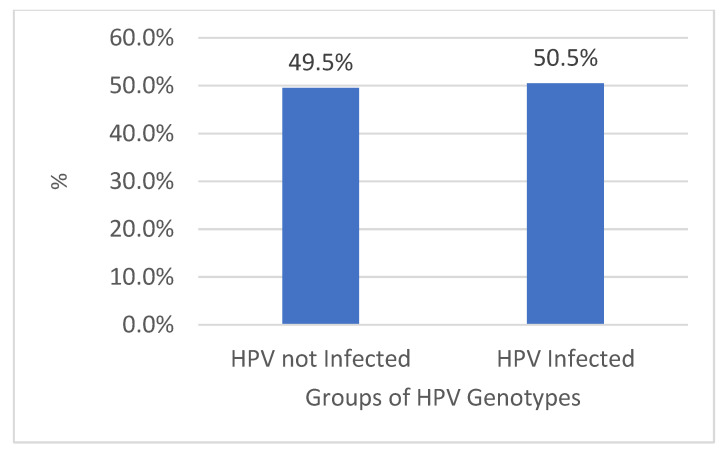
Percentage (%) of HPV-infected and non-infected patients.

**Figure 2 diagnostics-14-00968-f002:**
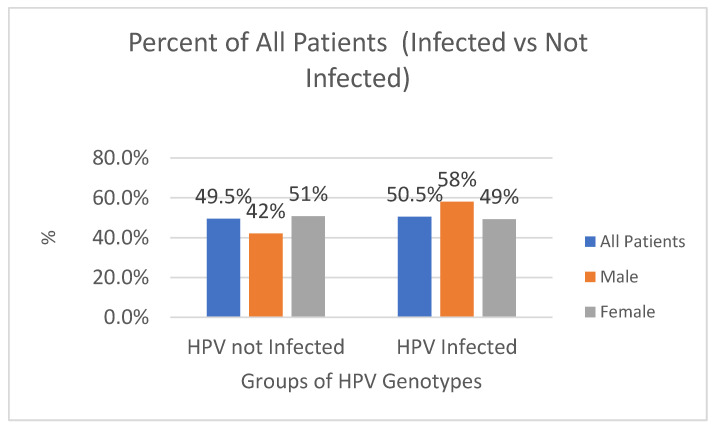
Percentage (%) of HPV-infected and non-infected patients by gender.

**Figure 3 diagnostics-14-00968-f003:**
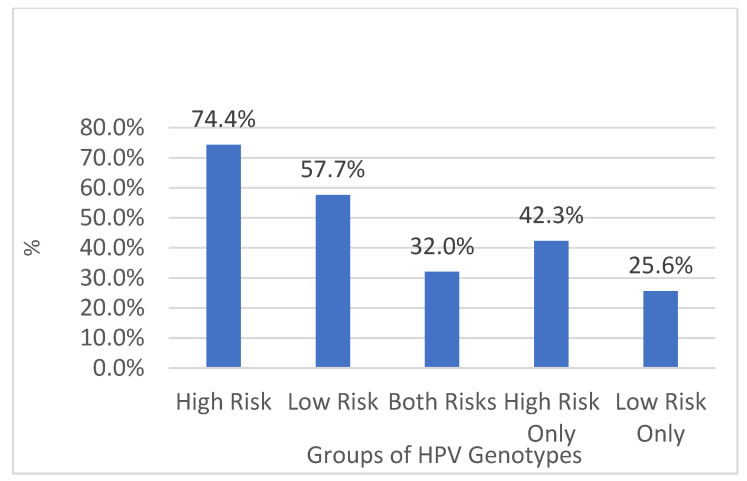
Percentage (%) of infected patients by HPV genotype groups.

**Figure 4 diagnostics-14-00968-f004:**
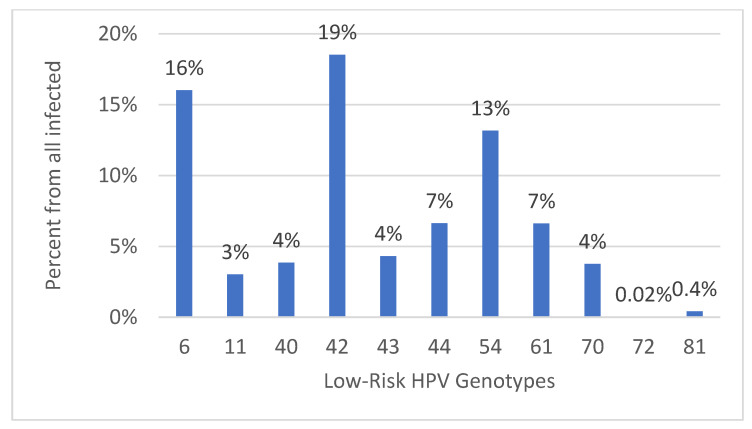
Percentage (%) of patients with low-risk HPV genotypes (*p* values > 0.05, 95% CI, 95%).

**Figure 5 diagnostics-14-00968-f005:**
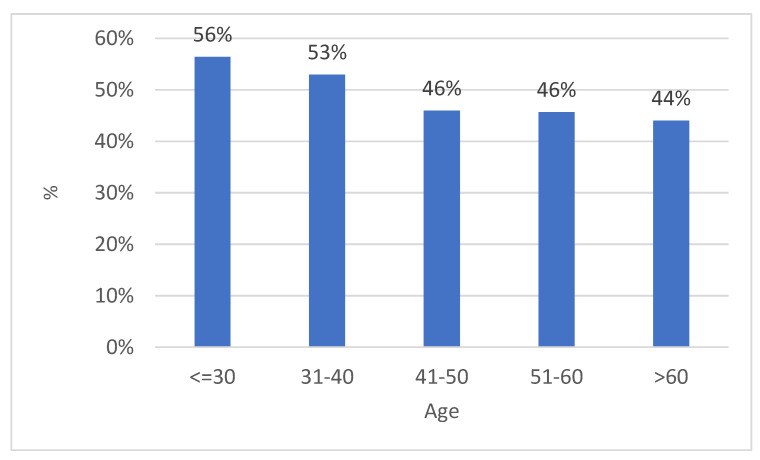
Percentage (%) of infected patients according to age group (*p* values < 0.05, 95% CI, 95%).

**Figure 6 diagnostics-14-00968-f006:**
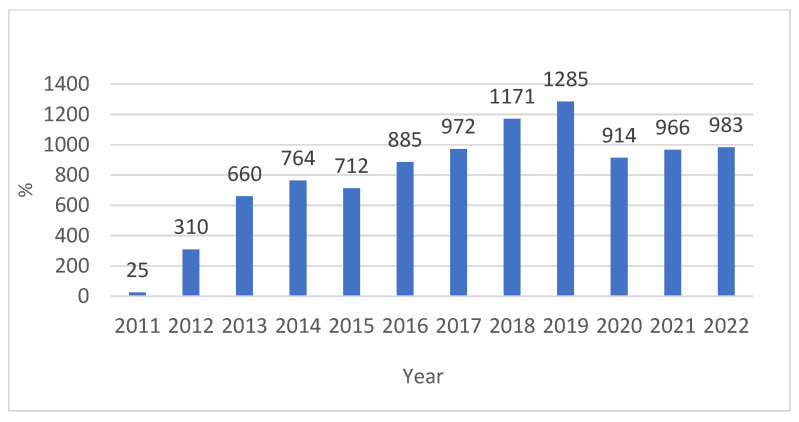
Total number of people undergoing HPV molecular testing over the years (2011–2022).

**Figure 7 diagnostics-14-00968-f007:**
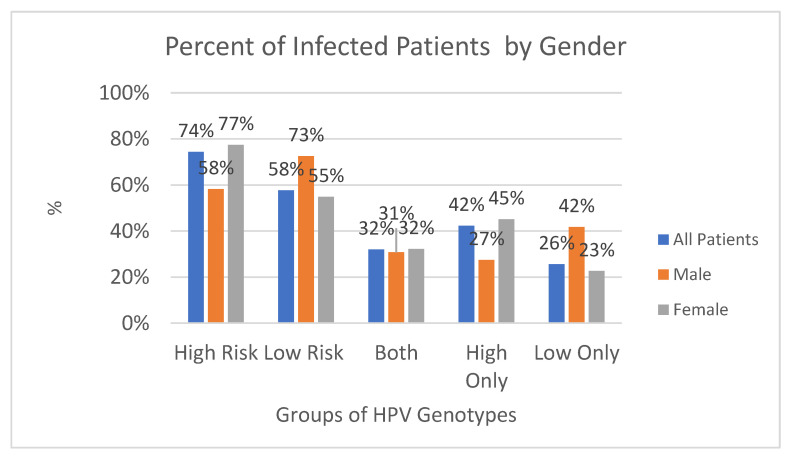
The total number of people undergoing HPV molecular testing over the years by gender (2011–2022). There was a lower percentage of males than females infected with high-risk genotypes. A greater percentage of males than females were infected with low-risk genotypes. An equal percentage of males and females were infected with both risk genotypes. A lower percentage of males than females were infected with high-risk genotypes only. A greater percentage of males than females were infected with low-risk genotypes only (*p* values < 0.05, 95% CI, 95%).

**Table 1 diagnostics-14-00968-t001:** Total number and percentage of HPV-infected and non-infected patients, also divided by gender.

	Total	Male		Female	
		*Infected*	*% of All M*	*Infected*	*% of All F*
**HPV not Infected**	4778	554	42%	4224	51%
**HPV Infected**	4869	764	58%	4105	49%
**Total**	9647	1318	100%	8329	100%

**Table 2 diagnostics-14-00968-t002:** Subdivision based on high and low risk genotypes considering the male and female groups, with absolute and relative percentages.

		All Genders		Male		Female	
	Genotypes	*Number*	*% All Genders*	*Number*	*% from Males*	*Number*	*% from Females*
HPV High-Risk Genotypes	High Risk	3621	74%	445	58%	3176	77%
HPV Low-Risk Genotypes	Low Risk	2807	58%	554	73%	2253	55%
HPV Both Risk Genotypes	Both	1559	32%	235	31%	1324	32%
HPV High-Risk Genotypes Only	High Only	2062	42%	210	27%	1852	45%
HPV Low-Risk Genotypes Only	Low Only	1248	26%	319	42%	929	23%
HPV Infected Total	Total Infected	4869	100%	764	100%	4105	100%

## Data Availability

Data is contained within the article.

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
