# Peer review of "Human Papillomavirus Carcinogenicity and the Need of New Perspectives: Thoughts from a Retrospective Analysis on Human Papillomavirus Outcomes Conducted at the Hospital University of Bari, Apulia, Italy, between 2011 and 2022"

_diagnostics, 2024, doi:10.3390/diagnostics14090968_

Round 1
Reviewer 1 Report
Comments and Suggestions for Authors
I congratulate the authors of the work and admire their efforts in gathering such a large study group.
I have a few comments:
1. The title seems long. I propose to shorten it, it is not understandable.
2. There are too many numbers in the abstract - I suggest including information about positive and negative numbers and the most common types. The abstract needs to be edited.
3. Keywords: too many of them, why adult stem cells; L1 and retrotransposition.
4. The introduction contains too much about the HPV virus as its structure, genetics, etc., and too little about genotyping and vaccination. This is a paper about genotyping.
I think it is unnecessary to write about other types of virus - line 76 or chlamydia - 1.3. HPV Relationship with Various Pathogens.
5. Inclusion and Exclusion Criteria
There is no information whether the patients were vaccinated or not - and this is very important. If vaccinated with what vaccine. Vaccination affects the recurrence of the disease and persistent infections.
Please find the citation.6. Figure 7 should not be in the discussion.
7.In the discussion, I propose to compare the genotyping results in other regions or countries, as we did in our work. For example like we do.
Przybylski, M.; Pruski, D.; Wszołek, K.; de Mezer, M.; Żurawski, J.; Jach, R.; Millert-Kalińska, S. Prevalence of HPV and Assessing Type-Specific HPV Testing in Cervical High-Grade Squamous Intraepithelial Lesions in Poland. Pathogens 2023, 12, 350.
8. Please consider work limitations before drawing conclusions.
9. There is too little about genotyping and too much about other topics such as virus, microbiota, chlamydia, etc. There is no table with information about the most common types of HPV, age, gender - it must be at work. Too many charts - one specific table would be enough. Do the authors have information about genetics from HSIL lesions - CIN2, CIN3 or cancers?
The work raises a very important topic regarding genotyping and the impact of the HPV virus on the development of pre-cancerous lesions and cervical cancer. Congratulations to the authors of the study group. I would like to ask you to respond to the above comments and take them into account. Please also write about the possible impact of HPV vaccination on the results of genotyping in the population.
Comments on the Quality of English Language
Author Response
1 The title seems long. I propose to shorten it, it is not understandable.
We thank the reviewer for the valuable suggestion, corrections have been made
2 There are too many numbers in the abstract - I suggest including information about positive and negative numbers and the most common types. The abstract needs to be edited.
We thank the reviewer for the valuable suggestion, corrections have been made
3 Keywords: too many of them, why adult stem cells; L1 and retrotransposition.
Thank you, adult stem cells and retrotransposition because we proposed the involvement of both in HPV infection, proliferation, and host cell gene modification towards CC as explained in the last part of our manuscript.
4 The introduction contains too much about the HPV virus as its structure, genetics, etc., and too little about genotyping and vaccination. This is a paper about genotyping.
Dear reviewer the main topic of this manuscript is a description of data collected during the last 11 years; this is a huge amount of data that we collected and put together in trying to give a sense of a valuable manuscript. No criteria nor vaccination were collected because this was a Government epidemiological study and we have to follow the rules.
5 I think it is unnecessary to write about other types of virus - line 76 or chlamydia - 1.3. HPV Relationship with Various Pathogens.
Due to the approach we have given to this article, this topic must necessarily be introduced. Many studies have confirmed the important role of different pathogens in HPV infection towards CC development. The C. trachomatis, K pneumoniae, E coli may enhance the risk of HPV infection and support viral persistence [39]. Chlamydia obstructs HPV-induced processes, such as mismatch repair (MMR), that maintain cellular and genomic integrity in coinfections.
7 Inclusion and Exclusion Criteria. There is no information whether the patients were vaccinated or not - and this is very important. If vaccinated with what vaccine. Vaccination affects the recurrence of the disease and persistent infections.
The reviewer is correct though all patients received at least 1 HPV vaccine we could not follow them as this was out of the purpose of the use and collection of the data.
In this article, beside the epidemiological view, we proposed a few line of discussion: (i), analysis and highlights on current understanding on virus/bacteria interaction during HPV infection and HPV-induced carcinogenesis; (ii), we highlighted the recent insights of the immune response against the virus, as well as the possible implications of retrotransposons during such virus/host cells interplay that may be translated into novel predictive and therapeutic approaches against HPV-induced cancers.
Please find the citation.
6 Figure 7 should not be in the discussion.
We thank the reviewer for the suggestion. change has been made.
7.In the discussion, I propose to compare the genotyping results in other regions or countries, as we did in our work. For example like we do.
We thank the reviewer for the suggestion. changes have been made.
8. Please consider work limitations before drawing conclusions.
We thank the reviewer for the suggestion. changes have been made.
Reviewer 2 Report
Comments and Suggestions for Authors
The manuscript: The Limitations of Current Beliefs on HPV Carcinogenicity and the Need to Include New Perspectives, Assumptions from a Retrospective Analysis on HPV Prevalence and Genotype Distribution among 9,647 patients in Bari, Apulia-Italy, between 2011 and 2022
Raffaele Del Prete, Daniela Nesta, Francesco Triggiano, Mara Lorusso, Stefania Garzone, Lorenzo Vitulano, Sofia Denicolo, Francesca Indraccolo, Michele Mastria, Luigi Ronga, Francesco Inchingolo, Sergey K Aityan, Kieu CD Nguyen, Toai Cong Tran, Ciro Isacco Gargiulo and Luigi Santacroce
Title: Title is too long. The number of patients examined and the period of their examination should be removed from the name. The title does not quite correspond to the essence of the article
Introduction: enough; represent the essence of the problem; does not require change
Methodology: meets the requirements of the journal and the branch of knowledge; does not require modification;
Literature: sufficient for the article and does not require additions
Article design: does not meet journal requirements
Conclusion: the article requires significant revision, after which it can be accepted for publication
The authors conducted a large study to study the prevalence of high- and low-risk HPV oncogenic risk for the period of time 2011-2022. Modern methods were used and interesting and important data were obtained.
General notes on the work:
1. The authors used statistical methods for processing the research results. However, these values are not given in the tables or figures. It seems to me that it is necessary to provide statistical data in the Research Results section.
2. In section 4 “Discussion”, there is no need to once again repeat the data obtained and described in the “Results” section and there is no need to refer to figures and tables that have already been presented in the previous section.
3. Why include Fig. in the “Discussion” section? 7? It must be moved to the “Results” section
4. The essence of the article is not clear – is this a literature review or your own data? Judging by the first part of the article, this is our own data. Therefore, they should be discussed in the “Discussion” section
5. Second part of the article, sections 4.1. “The Possible Role of Retrotransposition-L1 and Local Stem Cells in Increasing HPV 356 Proliferative Capacity” and 4.2. “The Role of Microbiota in the Carcinogenesis Process Induced by HPV in Adults and Elderly” – not relevant to the study results. Although the literature data presented are very important for understanding the significance of HPV in carcinogenesis.
Comments on the Quality of English LanguageAuthor Response
1 The authors used statistical methods for processing the research results. However, these values are not given in the tables or figures. It seems to me that it is necessary to provide statistical data in the Research Results section.
We thank the kindness and the valuable suggestions of the reviewer. We have edited following the suggestions
2 In section 4 “Discussion”, there is no need to once again repeat the data obtained and described in the “Results” section and there is no need to refer to figures and tables that have already been presented in the previous section.
We thank the kindness and the valuable suggestions of the reviewer. We have edited following the suggestions
3 Why include Fig. in the “Discussion” section? 7? It must be moved to the “Results” section
We thank the kindness and the valuable suggestions of the reviewer. We have edited following the suggestions
4 The essence of the article is not clear – is this a literature review or your own data? Judging by the first part of the article, this is our own data. Therefore, they should be discussed in the “Discussion” section
These data are our data. We have added our data discussion on the Discussion section
5 Second part of the article, sections 4.1. “The Possible Role of Retrotransposition-L1 and Local Stem Cells in Increasing HPV 356 Proliferative Capacity” and 4.2. “The Role of Microbiota in the Carcinogenesis Process Induced by HPV in Adults and Elderly” – not relevant to the study results. Although the literature data presented are very important for understanding the significance of HPV in carcinogenesis.
Our intention was also to cover something that is strictly related to HPV and its possible damage to cells, tissues, and organs therefore we added these chapters and paragraphs—the majority of patients who came for screening suffered from dysbiosis and inflammatory conditions, as well. For adults and the elderly, our data showed almost 50 to 50% equal infection and of course, the dysbiosis (elderly) and stem cells (younger) here may eventually explain these patterns; here wanted to give a different perspective that might explain the etiopathogenesis from HPV infection to development of neoplastic formation.
Reviewer 3 Report
Comments and Suggestions for Authors
Well structured article, the motivation of the study is interesting and I appreciate the recognition of the limits of the study
Author Response
We thank the reviewer for his/her appreciations
Reviewer 4 Report
Comments and Suggestions for Authors
The article reported the prevalence and genotype distribution of HPV in a Italian population. The author stated that their data showed similar results to previous studies.
1. The English language need moderate editing. At several places, the data were not presented clearly. For example, in the abstract, the overall prevalence of HPV was stated as 50.5% vs 49.5. However, the prevalence should be just one number. Also, there are many typos. For example, in the abstract, the total number of cases is supposed to be 9,647, but was written as 9.647. Similar mistakes are everywhere.
2. Could the author please point out what new data or conclusion they obtained through this study?
3. The introduction and the discussion are too long. In the discussion, the author proposed the potential role of L1, but I don't see how their dats is supporting that point.
4. In general, I think the authors should be more careful in their conclusion and what they can speculate based on that (for example, I don't see how the title is valid).
Comments on the Quality of English Language
need moderate editing
Author Response
1 The English language need moderate editing. At several places, the data were not presented clearly. For example, in the abstract, the overall prevalence of HPV was stated as 50.5% vs 49.5. However, the prevalence should be just one number. Also, there are many typos. For example, in the abstract, the total number of cases is supposed to be 9,647, but was written as 9.647. Similar mistakes are everywhere.
We thank the reviewer for the valuable critiques and suggestions. We edited.
2. Could the author please point out what new data or conclusion they obtained through this study?
Although this is a predominantly descriptive and epidemiological study, the data obtained reflect not only a fairly unique trend compared to other realities and latitudes, but also lead to a reflection that focuses on the infection of young people and adults, hypothesizing a process of HPV infection linked to both dysbiosis and stem cells and the retrotransposition mechanism. We have tried to better highlight these concepts.
3. The introduction and the discussion are too long. In the discussion, the author proposed the potential role of L1, but I don't see how their dats is supporting that point.
We tried to adjust following the reviewer's suggestions.
4. In general, I think the authors should be more careful in their conclusion and what they can speculate based on that (for example, I don't see how the title is valid).
Thank the reviewer for the suggestion, we have tried to adjust following his/her suggestions
Round 2
Reviewer 1 Report
Comments and Suggestions for Authors
Little corrections:
As observed by Przybylski and colleagues in their metanalysis, data need to be evaluated keeping an eye on different screening procedures, latitude, environments, social and food habits that may have influence either on HPV genotype itself or individuals microbiota.
The study certainly did not include vaccinated patients ? For sure ?
Congratulations on your work and recommends publications.
Author Response
As observed by Przybylski and colleagues in their metanalysis, data need to be evaluated keeping an eye on different screening procedures, latitude, environments, social and food habits that may have influence either on HPV genotype itself or individuals microbiota.
We thank you for the valuable suggestions we added the corrections.
The study certainly did not include vaccinated patients ? For sure ?
The data are not homogeneous therefore we did not include
Congratulations on your work and recommends publications.
Thanks a lot
Reviewer 2 Report
Comments and Suggestions for Authors
The manuscript: HPV Carcinogenicity, the Need of New Perspectives, Thoughts from a Retrospective Analysis on HPV Outcomes Hospital University of Bari, Apulia-Italy, Between 2011 and 2022
Raffaele Del Prete, Daniela Nesta, Francesco Triggiano, Mara Lorusso, Stefania Garzone, Lorenzo Vitulano, Sofia Denicolo, Francesca Indraccolo, Michele Mastria, Luigi Ronga, Francesco Inchingolo, Sergey K Aityan, Kieu CD Nguyen, Toai Cong Tran, Ciro Isacco Gargiulo and Luigi Santacroce
Title: corresponds to the content of the article
Introduction: enough; represent the essence of the problem; does not require change
Methodology: meets the requirements of the journal and the branch of knowledge; does not require modification;
Literature: sufficient for the article and does not require additions
Article design: does not meet journal requirements
Conclusion: the article meets the requirements of the journal and can be published without significant revision
The authors took into account all comments and made appropriate corrections.
There is one remark left regarding the value of p values.
In Fig. 1 "Percentage (%) of HPV infected and non-infected patients (p values < 0.05, 95% CI, 95%)".
In Fig. 2 "Percentage (%) of HPV infected and non-infected patients by gender (p values < 0.05,
95% CI, 95%)".
Maybe this is a mistake?
Data for HPV-infected and HPV-uninfected 49.5% and 50.5%. In my opinion, these differences are unreliable and p values must be >0.05
If this is the case, then the article can be published without significant changes

Author Response
In Fig. 1 "Percentage (%) of HPV infected and non-infected patients (p values < 0.05, 95% CI, 95%)".
In Fig. 2 "Percentage (%) of HPV infected and non-infected patients by gender (p values < 0.05, 95% CI, 95%)".
Maybe this is a mistake?
We thank the reviewer we have adjusted it.
it was
Data for HPV-infected and HPV-uninfected 49.5% and 50.5%. In my opinion, these differences are unreliable and p values must be >0.05
We thank the reviewer we have adjusted it.
Reviewer 4 Report
Comments and Suggestions for Authors
There are several places where "." and "," within numbers are wrongly placed in the results section.
Comments on the Quality of English Language
Minor English editing is still needed.
Author Response
There are several places where "." and "," within numbers are wrongly placed in the results section.
We thank the reviewer for the suggestion, we have added the correction